# Modulating the Surface Properties of Lithium Niobate Nanoparticles by Multifunctional Coatings Using Water-in-Oil Microemulsions

**DOI:** 10.3390/nano13030522

**Published:** 2023-01-28

**Authors:** Adrian Gheata, Alessandra Spada, Manon Wittwer, Ameni Dhouib, Emilie Molina, Yannick Mugnier, Sandrine Gerber-Lemaire

**Affiliations:** 1Institute of Chemical Sciences and Engineering, Ecole Polytechnique Fédérale de Lausanne, Group for Functionalized Biomaterials, 1015 Lausanne, Switzerland; 2Département de Chimie, École Normale Supérieure, PSL University, 75005 Paris, France; 3Université Savoie Mont-Blanc, SYMME, 74000 Annecy, France

**Keywords:** silanization, W/O microemulsion, surface functionalization, harmonic nanoparticles

## Abstract

Inorganic nanoparticles (NPs) have emerged as promising tools in biomedical applications, owing to their inherent physicochemical properties and their ease of functionalization. In all potential applications, the surface functionalization strategy is a key step to ensure that NPs are able to overcome the barriers encountered in physiological media, while introducing specific reactive moieties to enable post-functionalization. Silanization appears as a versatile NP-coating strategy, due to the biocompatibility and stability of silica, thus justifying the need for robust and well controlled silanization protocols. Herein, we describe a procedure for the silica coating of harmonic metal oxide NPs (LiNbO_3_, LNO) using a water-in-oil microemulsion (W/O ME) approach. Through optimized ME conditions, the silanization of LNO NPs was achieved by the condensation of silica precursors (TEOS, APTES derivatives) on the oxide surface, resulting in the formation of coated NPs displaying carboxyl (**LNO@COOH**) or azide (**LNO@N_3_**) reactive moieties. **LNO@COOH** NPs were further conjugated to an unnatural azido-containing small peptide to obtain silica-coated LNO NPs (**LNO@Talys**), displaying both azide and carboxyl moieties, which are well suited for biomedical applications due to the orthogonality of their surface functional groups, their colloidal stability in aqueous medium, and their anti-fouling properties.

## 1. Introduction

Nanoparticles (NPs) have attracted growing interest in the past decade for applications in biology and medicine, due to their unique properties [1]. Their size (ranging from one nanometer to a few hundred nanometers) matches that of biomolecules, such as proteins and nucleic acids, which is ideal for achieving long blood-circulation times [2]. Their high surface-area-to-volume ratio, combined with their ease of functionalization [3], makes them ideal materials to act as platforms that carry a high amount and diversity of molecular payloads and probes. Tailored chemical modification allows for the design of complex systems that are adaptable for a variety of applications, which explains why NPs have emerged as promising tools in biomedicine.

One major area of application for NPs is their use in drug and gene delivery. The aim of such drug-delivery systems (DDSs) is to increase the efficiency of treatments by improving various limiting properties of free drugs, including their solubility, stability in biological media, and biodistribution [4,5]. Owing to their potential for multi-functionalization, NP-based DDSs can also be decorated with targeting ligands (such as antibodies, nucleic acids, and peptides [6]) to increase the specificity of treatments. Another important area of interest is the use of NPs—more specifically, inorganic NPs that exhibit intrinsic imaging properties—as bioimaging probes [7]. A number of imaging modalities have been reportedly achieved by NP-based probes, such as magnetic resonance imaging (MRI) for deep tissue visualization—using magnetic iron oxide NPs [8,9] or lanthanide-doped NPs [10,11]—and optical imaging (OI) for high-sensitivity imaging of biological samples—achieved, for example, by quantum dots [12] or upconversion nanoparticles (UCNPs) [13]. Harmonic NPs (HNPs), such as bismuth ferrite (BiFeO_3_ [14]), barium titanate (BaTiO_3_ [15]), or niobates (NaNbO_3_, KNbO_3_, LiNbO_3_ [16]), have also recently appeared as highly efficient optical probes, due to several complementary advantages with respect to other optically active NPs [17,18]. These interesting properties include a coherent emission, the absence of bleaching/blinking when excited within the material transparency range, and wide spectral flexibility [19,20,21,22].

Regardless of the application targeted, the use of NPs in clinical practice is still restricted by their possible cell cytotoxicity and limited uptake efficiency [23,24,25]. Furthermore, nanocarriers can be rapidly removed from circulation by the reticuloendothelial system (RES) [26,27]. These factors are strongly influenced by the physicochemical properties of the studied NPs, such as their size, shape, surface composition, and superficial charge [28,29]. 

Current research efforts focus on tuning the properties of nanocarriers to address these obstacles and achieve better performances. Among different approaches, their surface modification and functionalization are emerging as powerful strategies to enhance their effectiveness and safety. It was proven that the surface coating of NPs can improve therapeutic efficiency [30], enhance biological properties [31], and achieve targeted and site-specific drug release [32]. Furthermore, surface-functionalized NPs are characterized by extended blood-circulation time, and have demonstrated the ability to evade immune cell recognition and uptake by the RES [33,34]. 

Several methodologies, based on the properties of NPs and the coating materials [35], can be implemented to achieve surface modification through non-covalent or covalent interactions. The first approach is simpler, but it is strongly affected by changes in pH and ionic strength [36]. On the other hand, covalent functionalization provides more flexible and robust systems. Furthermore, covalently conjugated systems are characterized by enhanced biodistribution and stability [35]. Polymers such as polyethylene glycol (PEG) have been extensively applied for the surface modification of nanocarriers. “PEGylation” shields the surface of NPs from opsonization, aggregation, and phagocytosis, resulting in a “stealth effect”, enabling the reduction of immunogenicity and prolonging blood-circulation times [37,38]. Additionally, exposure to PEG results in the production of anti-PEG antibodies that, in high concentrations, can reduce the clearance of PEGylated NPs.

Surface silanization is another promising approach for NP coating. Silica is known to be highly stable in aqueous media, biocompatible, and optically transparent. Furthermore, it is characterized by chemical inertness, controlled processability, and high porosity [39]. Therefore, silica provides protection from NPs’ degradation in their application environment [40]. As additional benefit, silica coatings of core NPs enable subsequent functionalization to tailor highly multifunctional NPs with designed stealth, targeting, and imaging properties. Several methodologies, such as mechanical stirring, heating, and a combination of both, have been investigated for the silanization of nanocarriers [41,42,43]. However, these procedures are laborious and require long processing times. Regarding the silica coating of multifunctional LNO NPs from homogeneous solutions, non-uniform the shell thickness and formation of multi-core NPs have been recently demonstrated [44]. Another type of process that has been extensively studied is sonication [45,46,47], but this methodology could lead to lower reproducibility.

Water-in-oil (W/O) microemulsions (MEs) (also known as reverse microemulsions), which consist of water droplets dispersed in an oil phase, were reported as promising candidates to act as nanoreactors for the silica coating of NPs [48,49,50,51,52]. MEs are thermodynamically stable dispersions of two immiscible fluids that are stabilized by the self-assembly of surfactant molecules at their interface [53]. As opposed to emulsions, in MEs both the micelle content and the surfactant molecules are constantly exchanged, promoting spatially confined chemical reactions that are involved in the NP coating. One of the key strengths of the use of reverse MEs is the better control over silica nucleation, which is limited to the nanodroplet core [54], allowing the formation of a uniform silica shell onto individual NPs. The use of ammonia as the aqueous phase in the presence of silica precursors (such as tetraethyl orthosilicate, TEOS [55]) leads to surface silica condensation through the known Stöber process [56,57]. Post-modification with organoalkoxysilane derivatives also allows the introduction of a second silica layer and of a variety of functional groups, such as carboxylic acid, azide, amino, or thiol moieties for subsequent functionalization. 

By tuning the ratio of oil/aqueous phases and the amount and type of surfactant, which are crucial parameters for the coating homogeneity, it is possible to control the droplet size and, therefore, the final size of the coated NP, showing high process versatility. Thus, the silica-coating processes of pre-synthesized NPs by the W/O ME method has found numerous applications, including the silanization of quantum dots [58,59], gold NPs [60,61], and magnetic NPs [62,63,64]. However, this appealing and promising methodology presents some challenges, such as the need for precise control over several parameters to achieve single core-shell NPs. Formation of core-free silica beads and multi-core particles are indeed two of the main obstacles encountered in this technique. A plausible reason is that the energy barrier for heterogeneous nucleation is lower than the one for homogeneous nucleation, resulting in competitive reactions. For this reason, it is vital to carefully tune the reaction parameters by tailoring the proportions of ammonia, oil phase, surfactant, TEOS, and NP cores. It is believed that the formation of single-core NPs is only achievable by matching the number of NPs with the number of droplets that accommodate them.

Herein, we report a controlled reverse ME method for the preparation of silica-coated core-shell harmonic NPs terminated with a variety of reactive moieties that are exploitable for further post-functionalization. The reaction parameters of the W/O MEs were optimized to match the number of NPs to the number of aqueous droplets that were initially prepared. The surface reactivity introduced after silanization was then exploited to conjugate a tailored unnatural peptide, thus demonstrating the flexibility of this nanoplatform toward post-functionalization pathways. Surface modifications at the different stages of the coating/functionalization processes were characterized using dynamic light scattering (DLS), Fourier transform infrared spectroscopy (FTIR), and transmission electron spectroscopy (TEM).

## 2. Materials and Methods

### 2.1. General Procedure for Coating LNO NPs Using a Reverse Micro-Emulsion Method

The procedure described below refers to the optimized conditions for the coating procedure. Span-20 (1.0 g) was dissolved in heptane (13 mL) and the solution was stirred at room temperature (rt) for 30 min. Aqueous NH_4_OH (28%, 50 µL) was added and the solution was stirred at rt for 3 h. The size of the resulting micelles was measured by dynamic light scattering (DLS). A suspension of LNO NPs in heptane (10 mg, 1 mL) was added and the mixture was stirred for 1 h. Tetraethyl orthosilicate (TEOS) (60 µL, 0.27 mmol) was added and the reaction mixture was stirred for 24 h at rt. Afterwards, a solution of (3-aminopropyl)triethoxysilane (APTES) derivative (APTES-COOH, synthesis procedure described in the Appendix A, or APTES-N_3_, synthesis procedure reported in [65], 0.03 mmol, 1.0 eq) in EtOH (300 mg/mL, 30 µL total volume) was added and the reaction mixture was stirred for 24 h at rt. The microemulsion was broken by the addition of MeOH (15 mL). The suspension was centrifuged (5 min, 4700 rpm) and the supernatant was discarded. The resulting NPs were washed with EtOH (1 mL, 5 times) and centrifuged, and the resulting **LNO@COOH** NPs or **LNO@N_3_** NPs were resuspended in EtOH (2 mL).

For DLS measurements, two aliquots of the **LNO@COOH** NPs or **LNO@N_3_** NPs suspension (5 µL each) were diluted with phosphate-buffered saline (PBS) 0.1× (1 mL, pH 7.4 and pH 3.0), and ultra-sonicated for 20 min. The diluted aliquots were analyzed with a Malvern NanoZ (Malvern Panalytical, Malvern, UK).

For FTIR characterization, an aliquot of the **LNO@COOH** NPs or **LNO@N_3_** NPs suspension in EtOH (0.4 mL, 1 mg/mL) was added to KBr (200 mg) and dried at 80 °C for 12 h before compression (2 min, 10 bars) to form a pellet. Samples were analyzed with a Nicolet 6700 FT-IR spectrophotometer (Thermo Fisher Scientific, Waltham, MA, USA).

### 2.2. Synthesis of Tri-Azidolysine (Talys)

#### 2.2.1. Glycine Loading on 2-Chlorotrityl Chloride Resin

The 2-chlorotrityl chloride (CTC) resin (0.5 g, 0.8 mmol, 1.0 eq) was loaded in a solid-phase extraction (SPE) tube, swollen for 30 min at rt in dry dichloromethane (DCM, 10 mL), then washed with dry DCM (5 mL, 3 times). Fmoc-Gly-OH (0.5 g, 1.6 mmol, 2.0 eq) was added to the tube, which was degassed by alternating argon/vacuum cycles (3 times). N,N-diisopropylethylamine (DIPEA, 0.55 mL, 3.2 mmol, 4.0 eq) diluted in dry DCM (5 mL) was added and the tube was shaken under argon for 1.5 h at rt. The resin suspension was filtered and washed with dimetyhlacetamide (DMA, 5 mL, 6 times). A mixture of DCM/MeOH/DIPEA (ratio 80/15/5, 15 mL) was added. The tube was shaken for 10 min at rt and filtered; this process was repeated twice. The resin was then washed with DMA (5 mL, 6 times).

Cleavage of the Fmoc group was performed by the addition of piperidine 25% in dimethylformamide (DMF, 10 mL) and the tube was then shaken for 5 min at rt, filtered, and washed with DMA (5 mL, 2 times) and DCM (5 mL, 2 times). Piperidine 25% (10 mL) was again added and the tube was shaken for 15 min at rt, then filtered. This step was repeated twice. The resin was washed with DMA (5 mL, 6 times), DCM (5 mL, 3 times), MeOH (5 mL, 3 times) and finally DCM again (5 mL, 6 times). The resin was left to dry overnight under vacuum.

#### 2.2.2. Azidolysine (Alys) Successive Coupling

The CTC@Gly resin was swollen for 30 min at rt in DCM (10 mL), then filtered and washed with DCM (5 mL) and DMA (5 mL). A solution of Fmoc-Alys-OH (synthesis described in the Appendix A; 0.47 g, 1.2 mmol, 1.5 eq), Hexafluorophosphate Azabenzotriazole Tetramethyl Uronium (HATU, 0.37 g, 0.96 mmol, 1.2 eq), and Hydroxybenzotriazole (HOBt, 0.13 g, 0.96 mmol, 1.2 eq) was prepared in DMA (5 mL). DIPEA (0.42 mL, 2.4 mmol, 3.0 eq) was added and the resulting solution was then added to the resin. The tube was shaken for 1.5 h at rt, then filtered and washed with DMA (5 mL, 6 times) and DCM (5 mL, 6 times). The progress of the coupling reaction was monitored by a Kaiser test.

Cleavage of the Fmoc group was performed by the addition of piperidine 25% in DMF (10 mL) and the tube was then shaken for 5 min at rt, filtered, and washed with DMA (5 mL, 2 times) and DCM (5 mL, 2 times). Piperidine 25% (10 mL) was again added and the tube was shaken for 15 min at rt, then filtered. This step was repeated twice. The resin was washed with DMA (5 mL, 6 times) and DCM (5 mL, 6 times), and the resin was left to dry overnight under vacuum.

This protocol was repeated two more times to obtain CTC@Gly-Alys-Alys-Alys.

#### 2.2.3. Final Glycine Coupling

The CTC@Gly-Alys-Alys-Alys resin was swollen for 30 min at rt in DCM (10 mL), then filtered and washed with DCM (5 mL) and DMA (5 mL). A solution of Fmoc-Gly-OH (0.36 g, 1.2 mmol, 1.5 eq), HATU (0.37 g, 0.96 mmol, 1.2 eq), and HOBt (0.13 g, 0.96 mmol, 1.2 eq) was prepared in DMA (5 mL). DIPEA (0.42 mL, 2.4 mmol, 3.0 eq) was added and the resulting solution was then added to the resin. The tube was shaken for 1.5 h at rt, then filtered and washed with DMA (5 mL, 6 times) and DCM (5 mL, 6 times). The progress of the coupling reaction was monitored by a Kaiser test.

Cleavage of the Fmoc group and resin drying were performed as described above (Section 2.2.2).

#### 2.2.4. Peptide Cleavage from Resin

The CTC@**Talys** resin was transferred to a flask and hexafluoro-2-propanol (HFIP) 25% in DCM (10 mL) was added. The resin was stirred at rt for 1 h, then diluted with HFIP (5 mL) and filtered. This step was repeated twice. The combined filtrates were concentrated under reduced pressure. The crude product was purified by preparative HPLC equipped with a C18 RP Waters OBD column. A linear gradient of solvent B (0.1% TFA in MeCN) over solvent A (0.1% TFA in water) rising linearly from 20% to 45% during t = 2.00–32.00 min was applied at a flow rate of 14.0 mL/min. Pure fractions containing the desired product were unified and lyophilized to afford **Talys** as a colorless fluffy material (0.38 g, 0.64 mmol, 80%). For the characterization of **Talys**, see the Appendix A.

### 2.3. Functionalization of Coated LNO NPs with Peptide Derivative

A suspension of **LNO@COOH** NPs in EtOH (1 mL, 1 mg/mL) was centrifuged (5 min, 4700 rpm). The supernatant was discarded and the NPs were resuspended in DMA (1 mL). A solution of pentafluorophenol (PFP-OH, 3.4 mg, 18 µmol, 6.0 eq) and 4-dimethylaminopyridine (DMAP, 0.074 mg, 0.6 µmol, 0.2 eq) in DMA was added and the suspension was stirred for 5 min at 0 °C. A solution of 1-ethyl-3-(3-dimethylaminopropyl)carbodiimide (EDC, 1.26 mg, 6.6 µmol, 2.2 eq) in DMA was added and the suspension was stirred for 5 min at 0 °C. The suspension was warmed to rt and left stirring for 3 h. The suspension was centrifuged (5 min, 4700 rpm) and the supernatant discarded. The solid residue was washed with DCM (1 mL, 3 times), centrifuged, and resuspended in DMA (1 mL). A solution of **Talys** (4.0 mg, 6.6 µmol, 2.2 eq) and DIPEA (50 µL, 287 µmol, 100 eq) in DMA was added and the suspension was stirred at 37 °C (reaction time dependent on derivative) for 3 h. The suspension was centrifuged (5 min, 4700 rpm) and the supernatant was removed. The solid residue was washed with a mixture of DCM–DMA (1:1, 1 mL, 2 times) and EtOH (1 mL, 2 times), centrifuged, and resuspended in EtOH (1 mL) to afford **LNO@Talys** NPs.

For DLS measurements, two aliquots of the **LNO@Talys** NP suspension (10 µL each) were diluted with PBS 0.1× (1 mL, pH 7.4 and pH 3.0) and ultra-sonicated for 20 min. The diluted aliquots were analyzed with a Malvern NanoZ.

For FTIR characterization, an aliquot of the **LNO@Talys** NPs suspension in EtOH (0.4 mL, 1 mg/mL) was added to KBr (200 mg) and dried in the oven overnight at 100 °C before compression (2 min, 10 bars) to form a KBr pellet.

STEM with EDX analysis was performed at the Interdisciplinary Centre for Electron Microscopy (CIME, EPFL, Lausanne, Switzerland) on an FEI Tecnai Osiris microscope (200 kV) and the samples were deposited on silica-free copper-carbon grids.

### 2.4. Protein Adsorption on LNO@Talys

A suspension of **LNO@Talys** or bare NPs in EtOH (1 mL, 0.5 mg/mL) was centrifuged (5 min, 4700 rpm). The supernatant was removed and the NPs were resuspended in 0.1 M MES buffer (1.5 mL, pH 6.0) and shaken for 30 min (700 rpm). The suspension was centrifuged (5 min, 4700 rpm) and the supernatant was removed. The solid residue was washed in 0.1 M MES buffer (1 mL, 2 times), centrifuged, and resuspended in sodium borate buffer (1 mL, 0.01 M, pH 8.8). A solution of neutravidin (Nav) in sodium borate buffer (0.5 mL, 50 µg/mL) was added and the suspension was shaken overnight at 4 °C (700 rpm). The NP suspension was centrifuged (5 min, 4700 rpm), and the supernatant was recovered. The solid residue was washed with sodium borate buffer (1 mL, 2 times) and PBS (1 mL, 1×, 2 times) before resuspension in PBS (1 mL, 1×). Supernatants and washing solutions were combined and subjected to ultrafiltration for quantification of unreacted neutravidin. After ultrafiltration, the sample was diluted in MilliQ to reach a maximum known protein concentration. The sample was centrifuged (5 min, 13,000 rpm) to remove traces of NPs. The concentration of unreacted neutravidin was quantified by micro-BCA assay. For the design of a typical calibration curve, standards with a variable neutravidin content were prepared. Each standard or sample replicate (150 µL) was pipetted into a 96-well-plate well. Micro-BCA assay working reagents (150 µL), made by mixing 25 parts of reagent A, 24 parts of reagent B and 1 part of reagent C, were added to each well and mixed thoroughly on a thermo shaker (500 rpm) for 30 s. The microplate was incubated at 37 °C for 2 h in the dark (500 rpm). After incubation, the microplate was cooled to rt and the absorbance at 562 nm was measured on a multi-plate reader (BioTek Synergy H1 multi-mode reader). The average 562 nm absorbance reading of the blank standard replicates were subtracted from the 562 nm reading of all other individual standards and sample replicates. The calibration curves, prepared by plotting the average blank-corrected 562 nm reading for each protein standard vs. their concentration in µg/mL, were used to determine the protein concentration of the sample. By indirect detection, the amount of neutravidin grafted on the surface of **LNO@Talys** NPs was calculated.

## 3. Results

### 3.1. Coating of LNO NPs by W/O Microemulsion

The use of reverse MEs was investigated in order to achieve a high control over the size and monodispersity of the resulting colloidal suspension of silica-coated LNO NPs. A two-step procedure was applied, involving the addition of TEOS for polycondensation at the NP surface through the Stöber process, followed by addition of the APTES derivative (APTES-COOH or APTES-N_3_), as shown in Figure 1.

In W/O MEs, small aqueous droplets dispersed in the oil phase act as “nanoreactors” which accommodate a single NP for subsequent silanization. The method enables the monitoring of droplet size for better control over the thickness of the silica layer, achieved through the thorough adjustment and tuning of parameters, such as the amounts of surfactants, the oil phase, the aqueous phase, and the silica precursors.

The stability and the efficiency of the reverse ME method strongly depend on the selection of the surfactant and the ratios of applied reagents. Screening of surfactants (Tween 80, Brij 35, AOT and Span 20) revealed that only Span 20 led to stable droplets over the time of the coating procedure. The use of cosurfactants, such as EtOH or octan-2-ol, led to a significant increase in the droplets’ size. The optimal volume ratio of oil to aqueous ammonia was empirically found to be 15 mL:50 µL (heptane:ammonia). Furthermore, the use of a more concentrated ammonia solution (28% aq) allowed an increase in the thickness of the silica coating layer. Coating procedures were performed at rt to avoid the evaporation of ammonia, due to its low boiling point. The silanization protocol lasted for 48 h and a slight increase of the size of the droplets was observed over the course of the reaction. However, only the initial size of the droplets needed to be finely tuned to accommodate the size of the bare LNO NPs.

The reverse ME method should avoid the formation of multi-core coated NPs or core-free silica NPs by matching the number of uncoated NPs (N_NPs_) with the number of stabilized droplets (N_droplets_), so that one droplet accommodates a single LNO HNP. N_NPs_ and N_droplets_ were defined as follows:(1)$$N_{\mathrm{NPs}}\;=\frac{m_{\mathrm{NPs}}}{m_{1\;\mathrm{NP}}}=\frac{m_{\mathrm{NPs}}}{d\;\times \frac{4}{3}\times \;\pi \;\times {\;r}_{\mathrm{NP}}^{3}\;}$$
where $m_{1\;\mathrm{NP}}\;$represents the mass of one NP, $m_{\mathrm{NPs}}\;$represents the total mass of NPs added in the reaction, d stands for the LNO density (d = 4.65 g/cm^3^), and $r_{\mathrm{NP}}\;$is the radius of the bare LNO NP (measured by DLS).
(2)$$N_{\mathrm{droplets}}\;=\frac{V_{\mathrm{aqueous}\;\mathrm{phase}}}{V_{1\;\mathrm{droplet}}}=\frac{V_{\mathrm{aqueous}\;\mathrm{phase}}}{\frac{4}{3}\times \;\pi \;\times {\;r}^{3}}$$
where$\;V_{\mathrm{aqueous}\;\mathrm{phase}}$ is the volume of aqueous ammonia added in the reaction and r is the radius of one droplet (measured by the DLS size distribution in number during the procedure, i.e., before addition of LNO NPs). The ammonia droplets were measured and had a radius of 85 nm after 3 h of stabilization (see the Appendix A), but their size was observed to increase over time. After 7 h, their size reached 100 nm in radius and went up to 130 nm after 20 h. If this last value is taken for the radius of the droplets (halfway through the silanization procedure), considering that bare LNO NPs have an average radius of about 50 nm (measured by DLS), the amount of NP required to match the number of ammonia droplets corresponds to 10 mg for the conditions described above.

Bare LNO NPs were prepared at a low hydrolysis rate of alkoxide precursors with a solvothermal route already described [16] (see the Appendix A, for characterization of bare LNO NPs). Solution-based approaches are indeed more adequate to prepare non-centrosymmetric oxides at the nanoscale, such as lithium niobate from alkoxide precursors [66,67].

The first attempts at silanization of LNOs using the W/O ME method were performed with the APTES-COOH derivative. Several reaction parameters were screened to optimize the coating conditions and to achieve monodispersity of the resulting **LNO@COOH** NP suspension. After stabilization of the NP/ME mixture for 3 h, TEOS was added and left to react for 24 h. Addition of APTES-COOH was carried out after dissolution in EtOH at a concentration of 300 mg/mL, to facilitate handling and ease its transfer to the aqueous droplets, due to EtOH acting as a cosurfactant. The silanization procedure afforded **LNO@COOH** NPs with an average hydrodynamic diameter of 150 to 180 nm for the two batches described (**LNO@COOH (2)** and **(1)**, respectively) and a high monodispersity (PDI < 0.1) of the resulting colloidal suspension in PBS buffer. The presence of surface hydrophilic groups also resulted in large pH-dependent variations of the surface charge (on average from −43 mV at pH 7.4 to −12 mV at pH 3.0, as shown in Table 1), consistent with the presence of carboxylates at the NP surface.

Modification of the reactive functionalities introduced at the surface of LNO NPs was then achieved by applying the coating procedure in the presence of APTES-N_3_. The resulting **LNO@N_3_** NPs were analyzed by DLS (Table 1) in EtOH and PBS 0.1×. In EtOH, a significant increase in size was observed, compared to bare LNO NPs, and monodispersity of the suspension was preserved (PDI = 0.07). In the buffered aqueous solution, **LNO@N_3_** NPs showed signs of agglomeration (large hydrodynamic diameter and PDI > 0.2), which was attributed to the high density of hydrophobic chains introduced at the NP surface. The zeta potentials measured at pH 7.4 and 3.0 indicated a shift toward more positive values in both cases, which is consistent with a shielding of the negative surface charge of the metal oxide core by the coating layer.

Fourier transform infrared spectroscopy (FTIR) was used to assess the efficiency of the coating method and to confirm the presence of the silica layer, as well as COOH or N_3_ moieties at the NP surface (Figure 1). All spectra were normalized with respect to the O-Nb-O asymmetric and symmetric bands (720 and 650 cm^−1^, respectively) [68,69,70]. **LNO@COOH** NPs presented peaks related to the linear stretching of the O-Si-O bond at 1080 cm^−1^ and a band characteristic of the Si-CH_2_ symmetric bending at 1200 cm^−1^, indicating the formation of the silica coating layer. The availability of surface carboxylic moieties was evidenced by the presence of a broader band for O-H stretch (~3300 cm^−1^), a sharp band for C=O stretch (~1640 cm^−1^), as well as C-O stretch (~1320 cm^−1^). In the case of **LNO@N_3_**, NPs especially displayed an intense band for the characteristic N=N=N stretch (~2100 cm^−1^). The presence of these intense peaks provided evidence for the efficiency of the silanization procedure and confirmed the presence of surface reactive carboxylic or azido moieties for downstream functionalization.

The morphology of **LNO@COOH** and **LNO@N_3_** NPs was further investigated by transmission electron microscopy (TEM). Formation of a thin but dense silica coating layer around the crystalline core was demonstrated in the case of **LNO@COOH** NPs (Figure 2a,b, average thickness of the coating layer measured at 7 nm). However, the silica coating of **LNO@N_3_** NPs was not as homogeneous (Figure 2c,d). The formation of silica clusters was observed in some regions of the NP surface, while part of the inorganic cores was covered with a very thin silica layer. Energy-dispersive X-ray (EDX) elemental analysis showed an atomic ratio of silicon to niobium of 10% for **LNO@COOH** NPs and 33% for **LNO@N_3_** NPs. The high value observed for **LNO@N_3_** NPs resulted from the presence of silica clusters and did not reflect a homogeneous availability of the reactive azido group at the NP surface. We suspected that the hydrophobic character of the APTES-N_3_ silanization reagent was responsible for the poor quality of the resulting silica coating.

The silanization procedure using the W/O ME method afforded coated LNO NPs displaying surface reactive moieties, available for further functionalization that can increase the diversity of molecular components at the NPs’ surface and modulate their behavior in a physiological medium. In view of their favorable properties in an aqueous medium, **LNO@COOH** NPs were selected to explore post-functionalization procedures.

### 3.2. Functionalization of LNO@COOH NPs with Peptide Derivative

As discussed in Section 3.1, **LNO@N_3_** NPs displayed monodisperse suspension in an alcoholic solution and a high density of surface azido groups. However, their tendency to agglomerate in an aqueous solution and the uneven silanization of **LNO@N_3_** NPs led us to explore an alternative strategy for the introduction of azido groups, while ensuring monodisperse suspension in aqueous medium (PDI < 0.2). Conjugation of **LNO@COOH** NPs with a branched peptide containing multiple azide-bearing side chains was envisaged to provide the simultaneous presence of azido and carboxylate surface orthogonal functionalities and to improve the hydrophilic character of the functionalized NPs for enhanced colloidal stability in an aqueous medium. The peptide structure design was based on repeating azide-modified lysine units (azidolysine = Alys, synthesis described in the Appendix A). As the number of Alys units could be easily modified, the length of the peptide backbone could be tuned to exhibit various azide-to-carboxylate ratios. The peptide described herein (tri-azidolysine = **Talys**, Figure 3) contains three Alys units, and was capped with one glycine unit at each end to reduce the steric hindrance on the amine and carboxylic acid groups, thus increasing their reactivity.

#### 3.2.1. Solid-Phase Synthesis of Talys

**Talys** was produced by solid-phase peptide synthesis (SPPS, Figure 2), using Fmoc-protected amino acids as building units. Fmoc-protected glycine was loaded onto 2-chlorotrityl chloride (CTC) resin, followed by deprotection of the terminal amine with piperidine 25% in DMF. Fmoc-protected Alys was then successively coupled and deprotected three times on CTC@Gly. Finally, the last glycine unit was added through coupling and Fmoc deprotection, after which **Talys** was cleaved from the resin using HFIP 25% in DCM and purified by preparative HPLC.

#### 3.2.2. Conjugation of Talys to LNO@COOH NPs

**Talys** was grafted to **LNO@COOH** by amide bond coupling (Figure 3). The selected conditions started with the activation of the carboxylic acid groups present on the surface using PFP-OH in the presence of EDC and DMAP, generating the PFP ester. The excess of coupling agents was then washed by centrifugation and the activated **LNO@PFP** NPs were treated with **Talys** and DIPEA to afford **LNO@Talys** NPs that were characterized by DLS.

As evidenced by the measurements of their size and PDI, **LNO@Talys** NPs exhibited a remarkable ability to suspend in both organic (EtOH) and aqueous media (PBS), resulting in the formation of homogeneous colloidal suspensions (PDI of 0.10 and 0.06 in EtOH and PBS 0.1×, respectively) (Table 2). The slight increase in size and very similar zeta potentials compared to **LNO@COOH** NPs are consistent with surface conjugation to a small peptide.

**LNO@Talys** NPs were further characterized by FTIR (Figure 4), which provided evidence for the introduction of azido groups, corresponding to their characteristic band at 2100 cm^−1^ (though smaller in intensity as compared to **LNO@N_3_** NPs). The signals attributed to the silanization (1640, 1200 and 1080 cm^−1^) were also mostly identical between **LNO@COOH** and **LNO@Talys** NPs, indicating that the coupling procedure was not detrimental to the integrity of the silica shell.

The morphology of **LNO@Talys** NPs was further investigated by transmission electron microscopy (TEM), confirming that the dense and homogeneous silica shell around the crystalline core formed during the silanization step was maintained (Figure 5, average thickness of the coating layer was measured at 7 nm, similar to the value measured for **LNO@COOH** NPs). Energy-dispersive X-ray (EDX) elemental analysis showed an atomic ratio of silicon to niobium of 12%, similar to the one described for the starting coated NPs (Section 3.1).

#### 3.2.3. Protein Adsorption on LNO@Talys

Protein adsorption shields the original surface properties and confers new biological properties to NPs. Therefore, pharmacokinetic and toxicological profiles would be drastically changed after formation of a protein corona upon injection, causing a variation in biological fate and efficacy [71]. Processes such as PEGlyation and zwitterionization are known strategies to reduce or eliminate the formation of protein corona [72,73]. However, it was recently demonstrated that functionalizing or precoating the NPs’ surface with proteins or peptides reduces the adsorption of serum proteins present in the blood, enhancing the evasion from the immunosurveillance [74,75]. In line with these results, micro-BCA protein quantification assays using neutravidin as the model protein demonstrated that **LNO@Talys** NPs allowed a drastic reduction of protein adsorption, compared to bare LNO NPs. Quantification experiments showed a significant decrease in the amount of neutravidin adsorbed on the NP surface (Figure 6, *p* < 0.05, Student *t*-test). This observation confirms the high potential of **LNO@Talys** NPs for use as a platform for biomedical applications.

## 4. Discussion

Silanization is a common approach to passivate the surface of otherwise toxic inorganic NPs, and it can be used to introduce chemical handles at their surface for post-functionalization. This strategy allows for the design of complex systems, combining the inherent properties of the NP core with the ones of the various components that can be grafted. Although silica coating of NPs has already been extensively described, there is still a need for robust protocols that can produce monodisperse NPs and that can be easily adapted for the introduction of various functional groups, depending on the targeted post-functionalization strategy. The results presented herein show strategies for the silanization of LNO metal oxide NPs using W/O ME technology, which is amenable to the introduction of either carboxylic acid or azide moieties at the NP surface.

The silanization protocol was first demonstrated with the introduction of carboxylates for downstream functionalization through amide bond formation. **LNO@COOH** NPs displayed monodisperse colloidal suspension in an aqueous medium, which is an important parameter for further potential biological applications, particularly for systemic administration where the formation of aggregates should be avoided to prevent undesirable side effects. Under similar conditions, **LNO@N_3_** NPs were produced and formed monodisperse colloidal suspension in an alcoholic solution (EtOH). However, the high density of surface aliphatic chain resulted in a clear tendency to aggregate in an aqueous medium, and the silica coating was not homogeneous.

The introduction of azido groups is still appealing, due to the ease of functionalization by bioorthogonal click cycloaddition [76,77,78], but the poor colloidal stability of **LNO@N_3_** NPs in water was an important limiting factor for further applications. To overcome this issue, the azide-bearing peptide **Talys** was synthesized and conjugated to **LNO@COOH** NPs, using PFP esters as activated intermediates. The resulting **LNO@Talys** NPs showed monodispersity in both ethanol and PBS, confirming its dispersibility in both media. The presence of surface carboxylic acid moieties and azido groups was confirmed by FTIR analysis. The propensity of **LNO@Talys** to adsorb proteins by electrostatic interactions was also evaluated. Compared to bare LNO NPs, **LNO@Talys** NPs exhibited a clearly lower protein adsorption. The properties of **LNO@Talys** NPs, combining monodisperse colloidal behavior in an aqueous medium and minimal protein absorption capacity, offer promising perspectives for potential future biological applications. Furthermore, the fact that these NPs also display orthogonal azido and carboxylic acid groups makes them excellent candidates for multi-functionalization with chemical components of varying properties, paving the way for the design of complex nanoplatforms.

## Data Availability

The data presented in this study are openly available in Zenodo.

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
