# Peer review of "Modulating the Surface Properties of Lithium Niobate Nanoparticles by Multifunctional Coatings Using Water-in-Oil Microemulsions"

_nanomaterials, 2023, doi:10.3390/nano13030522_

Round 1
Reviewer 1 Report
Nanomaterials
Editor
Dear Editor,
The results presented in the manuscript could be interesting for readers. Only minor improvement could be done.
1. The title. I suggest to avoid any abbreviations in the title and changed the title to “Modulating the Surface Properties of Lithium Niobate Nanoparticles by Multifunctional Coatings Using Water-in-Oil Microemulsions”.
2. The Introduction part is very comprehensive, however, the selection of synthesis of namely LiNbO3 should be also stressed in the Introduction.
3. Figure 1. The sentence “All spectra were normalized with respect to the O-338 Nb-O asymmetric and symmetric bands (720 and 650 cm-1, respectively)” needs some citation. The intensive absorption bands at 1080 cm-1 and at about 450 cm-1 observed in FTIR spectra are not attributed to any specific vibrations. In the text authors mentioned “and O-H bend (~ 960 cm-1) bands”, however, it is difficult to find this bands in FTIR spectra. Similar comments are on Figure 4.
The manuscript could be suitable for publication in this journal after minor revision.
Reviewer 2 Report
The manuscript submitted for review describes a recipe for creating functionalized nanoparticles designed for targeted delivery of active substances into the body. The authors have developed a technology for the synthesis of core-shell nanoparticles with a core based on Lithium Niobate. In this case, the core is covered with a thick layer of silica. The given spectra convincingly demonstrate the complete closure of the core. At the same time, SEM data showed that the core is covered with only a thin layer of silica. Nevertheless, it seems that the Lithium Niobate core is completely isolated from functional groups. In this regard, the reviewer has only one question. How necessary is it to use nanoparticles with a core of exactly this composition? Perhaps it would be worth paying attention to more accessible and less dangerous materials (Li) for humans. In addition, there are no conclusions in the text.
Round 2
Reviewer 2 Report
After reading the revised version of the manuscript submitted for review, I have to repeat my previous comment. The research was certainly carried out at the highest scientific level and using modern laboratory methods. The authors of the study have achieved great success in creating a silica shell around the core of lithium niobate. The question is, how necessary is the use of this substance in the core of the nanoparticle? Indeed, lithium niobate has a number of unique optical properties, which makes it possible to study the obtained nanoparticles and ligand complexes based on them by optical methods. However, the question arises as to how applicable these objects are in practice? If these are model systems that only demonstrate the capabilities of the method developed by the authors of the manuscript, then this should be indicated in the introduction to the article. If the obtained core-shell nanoparticles still have the potential for use in medicine, then it is necessary to note how their unique optical properties will manifest themselves inside the body. What will be the light source inside almost opaque human or animal tissues and organs? How will the radiation emitted by nanoparticles be recorded? Without an answer to these questions, the article cannot be published.